# COMSUC: A web server for the identification of consensus molecular subtypes of cancer based on multiple methods and multi-omics data

**Song He**[1]☯, **Xinyu Song**[2]☯, **Xiaoxi Yang**[1,3]☯, **Jijun Yu**[4]☯, **Yuqi Wen**[1], **Lianlian Wu**[1], **Bowei Yan**[1], **Jiannan Feng**[4], **Xiaochen Bo**[1]*

**1** Department of Biotechnology, Beijing Institute of Radiation Medicine, Beijing, P.R.China, **2** Key Laboratory of Biomedical Engineering and Translational Medicine, Ministry of Industry and Information Technology, Chinese PLA General Hospital, Beijing, P.R.China, **3** Experimental Center, Beijing Friendship Hospital, Capital Medical University, Beijing, P.R.China, **4** State key Laboratory of Toxicology and Medical Countermeasures, Beijing Institute of Pharmacology and Toxicology, Beijing, P.R.China

☯ These authors contributed equally to this work.
* boxc@bmi.ac.cn, boxiaoc@163.com

**Data Availability Statement:** All data files are available from the TCGA database, ICGC database and TARGET database (doi: 10.5114/wo.2014. 47136, doi: 10.1038/s41587-019-0055-9, https:// ocg.cancer.gov/programs/target).

## Abstract

Extensive amounts of multi-omics data and multiple cancer subtyping methods have been developed rapidly, and generate discrepant clustering results, which poses challenges for cancer molecular subtype research. Thus, the development of methods for the identification of cancer consensus molecular subtypes is essential. The lack of intuitive and easy-to-use analytical tools has posed a barrier. Here, we report on the development of the COnsensus Molecular SUbtype of Cancer (COMSUC) web server. With COMSUC, users can explore consensus molecular subtypes of more than 30 cancers based on eight clustering methods, five types of omics data from public reference datasets or users' private data, and three consensus clustering methods. The web server provides interactive and modifiable visualization, and publishable output of analysis results. Researchers can also exchange consensus subtype results with collaborators via project IDs. COMSUC is now publicly and freely available with no login requirement at http://comsuc.bioinforai.tech/ (IP address: http://59.110. 25.27/). For a video summary of this web server, see S1 Video and S1 File.

## Author summary

A number of methods have been developed for omics data-based subtyping, which has been widely accepted as a relevant source of cancer classification. However, discrepant results hamper the translational and clinical utility of these methods. In this study, we have developed the COnsensus Molecular SUbtype of Cancer (COMSUC) web server to provide a user-friendly tool for integrating discrepant clustering results based on multiple platform, multiple omics data and multiple methods. COMSUC provides powerful

**Funding:** This work was supported by National Key R&D Program of China [2016YFC0901600]. The funders had no role in study design, data collection and analysis, decision to publish, or preparation of the manuscript.

**Competing interests:** The authors have declared that no competing interests exist.

support to users to decipher the cancer Consensus Molecular Subtypes (CMSs), a consensus classification system integrating different clustering results.

This is a *PLOS Computational Biology* Software paper.

## Introduction

With rapid progress in high-throughput technologies, parallel acquisition of multi-omics data for cancer is becoming less expensive, resulting in the accumulation of large-scale multidimensional cancer databases [e.g., Therapeutically Applicable Research To Generate Effective Treatments (TARGET), https://ocg.cancer.gov/programs/target] [1–4]. Several methods have been developed for omics data–based subtyping, which has been accepted widely as a relevant form of cancer classification [5–7]. However, discrepant results compromise the translational and clinical utility of these methods. Guinney J, et al. refers that different colorectal cancer classification methods can only identify subtype based on microsatellite instability and highly expressed mesenchymal genes, but failed to achieve consistency for other subtypes [8]. And they applied a network-based algorithm to examine consistency among six independent colorectal cancer classification systems, and to merge them into four consensus molecular subtypes (CMSs). Thus, methods and tools for the identification of cancer CMSs through the integration of discrepant clustering results from multi-methods and multi-omics data are valuable resources for researchers.

Several programing packages have been developed for this purpose. For example, the "ConsensusClusterPlus" and "MCL" R packages enable the integration of multiple clustering results using resampling and network-based approaches [9,10]. However, they are implemented without a user interface of consensus clustering algorithms, which may be too complex for most researchers. Additionally, previously published web applications, such as ICM and ArrayMining, focus on subtyping based only on single clustering methods or omics data types [11,12]. For example, ICM, a web server developed by our group in 2016, can integrate only multi-omics data, but not clustering results generated by different methods. Despite efforts to date, an intuitive user interface permitting convenient analysis of consensus clustering results by biologists and clinicians remains lacking.

In this research, we developed the COnsensus Molecular SUbtype of Cancer (COMSUC) web server, which provides a user-friendly tool for the integration of discrepant cancer clustering results generated by multiple methods and multi-omics data from multiple platforms. COMSUC is now available at http://comsuc.bioinforai.tech/ (IP address: http://59.110.25.27/). It is free and open to all users and has no login requirement. COMSUC deploys eight typical clustering methods and three consensus clustering approaches. It can be used to identify CMSs of more than 30 cancers. COMSUC data are from three cancer research projects [The Cancer Genome Atlas (TCGA), the International Cancer Genome Consortium (ICGC), and TARGET] [2,4]; users' private datasets can also be entered. We hope that COMSUC will provide powerful support for users' efforts to decipher cancer CMSs, and for the development of a consensus classification system integrating different clustering results.

## Design and implementation

### Implementation

COMSUC has two main components: a user interface and computational services. The user interface permits interactive analysis and visualization of results, and was written using the

iView framework for HTML, CSS, and JavaScript. "Cytoscape.js" (a JavaScript plug-in) is used to show the interactive clustering network. The D3 library of JavaScript is used to illustrate heatmaps. The computational services are provided by the back-end server using the Egg.js framework for Node.js. COMSUC uses R to implement algorithms and MySQL for the storage and management of application data.

## Data sources

Although the COMSUC web server allows users to upload their own data for analysis, it was also designed to provide intuitive analysis of publicly available datasets from TCGA, the ICGC, and TARGET. The dataset contains five types of omics data for 14,954 patient samples: mRNA, miRNA, DNA methylation, copy number variation, and reverse-phase protein array (RPPA) data. These data were downloaded from the University of California at Santa Cruz Xena browser (https://xenabrowser.net/datapages/). The following preprocessing steps were performed to improve dataset quality:

i. patient samples lacking omics data were filtered out;

ii. patient samples with data for >20% features missing, and features for which >20% patient samples were missing, were filtered out;

iii. other missing data were filled in using average imputations.

## Feature selection and batch effect correction

Features were pre-selected for each type of omics data. mRNA features with the top 1,500 and miRNA features with the top 647 median absolute deviations were selected. DNA methylation and copy number variation features with the top 1,500 maximum standard deviations were selected. All RPPA features were included in the subsequent steps. The feature selection thresholds and methods are referred to the Broad Institute GDAC firehose analyses workflow (http://gdac.broadinstitute.org/Analyses-DAG.html).

For users' uploaded private data, COMSUC applies the same feature selection algorithm before clustering if the feature dimension of private data is more than that of preprocessed public data in COMSUC. For users wishing to identify CMSs using their private datasets and the public datasets, COMSUC uses the "ComBat" function in the "sva" R package to perform batch effect correction. Then, feature selection is performed before clustering [13].

## Clustering algorithms

COMSUC integrated eight typical clustering methods implemented using R packages (Table 1). Detailed information about these methods is provided on the "Manual" page of the COMSUC website.

The optimal clustering number was evaluated using different clustering performance assessment indexes. Evaluation for the Kmeans, Hclust, SOM, and APclust methods was performed using the ch, db, silhouette, and dunn indexes, implemented with the "NbClust" function in the "NbClust" R package (S2 File) [14]. The best clustering number for iKmeans and iHclust was evaluated using the empirical cumulative distribution function, implemented with the "ecdf" function in the "ConsensusClusterPlus" R package [9]. For the NMF method, the cophenetic correlation coefficient index was implemented with the "nmf" function in the "NMF" R package [15]. For the Spectralclust method, the eigen-gap index was implemented with the "estimateNumberOfClustersGivenGraph" function in the "SNFtool" R package [16].

**Table 1. Clustering algorithms implementation.**

| Clustering Methods in COMSUC | R Package | R Function |
| --- | --- | --- |
| Kmeans | stats | kmeans |
| iKmeans | ConsensusClusterPlus | ConsensusClusterPlus |
| Hclust | stats | hclust |
| iHclust | ConsensusClusterPlus | ConsensusClusterPlus |
| NMF | NMF | nmf |
| SOM | kohonen | somgrid, supersom |
| Spectralclust | SNFtool | spectralClustering |
| APclust | apcluster | apcluster, apclusterK |

### Cancer CMS identification algorithm

COMSUC deploys three consensus clustering algorithms, MCL, COCA, and SuperCluster, to integrate clustering results generated by different clustering methods and multi-omics data. These methods are successfully used in the identification of cancer subgroups or pan-cancer groups based on multiple methods and multi-omics data consensus clustering [8,17–19]. The MCL method was implemented with the "mcl" function in the "MCL" R package [10]. The optimal clustering number was evaluated using the weighted silhouette coefficient, implemented with the "WeightedCluster" R package [20]. The COCA and SuperCluster method were implemented with the "ConsensusClusterPlus" function in the "ConsensusClusterPlus" R package [9]. The best clustering number for COCA and SuperCluster was evaluated using the empirical cumulative distribution function, implemented with the "ecdf" function in the "ConsensusClusterPlus" R package [9].

### Survival analysis and subtype signature discovery

COMSUC uses the Kaplan-Meier method to perform survival analysis, implemented with the "survival" R package[21]. Subtype signatures are examined using the "pamr" R package [22].

## Results

### Overview of COMSUC

COMSUC can be accessed with most web browsers, including Mozilla Firefox, Safari, Google Chrome, and Microsoft Edge. For the best visualization, Google Chrome is recommended.

COMSUC identifies CMSs by three steps (Fig 1A). First, it uses eight typical clustering methods [K-means (Kmeans), iterative K-means (iKmeans), hierarchical clustering (Hclust), iterative hierarchical clustering (iHclust), non-negative matrix factorization (NMF), self-organizing maps (SOM), spectral clustering (Spectralclust), and affinity propagation clustering (APclust)] to divide samples into clusters. Next, COMSUC uses one of three consensus clustering methods [the network-based Markov clustering (MCL), Cluster-of-Cluster Assignments (COCA), and SuperCluster], to integrate these clustering results. Then, COMSUC uses the consensus clustering algorithm for further integration of the clustering results based on omics data.

On the "Analysis" page of the website, users can choose to analyze one of 35 cancer types (Fig 1B). Next, users have three choices about input data: they can upload their private data, choose one of three public cancer data sources (TCGA, ICGC, and TARGET), or do both. Then, users choose any of five omics data types, any of eight typical clustering methods, or any of three consensus clustering methods. Finally, by running the analysis application, users can interactively explore consensus results by networks, heatmaps, survival curves, and CMS-

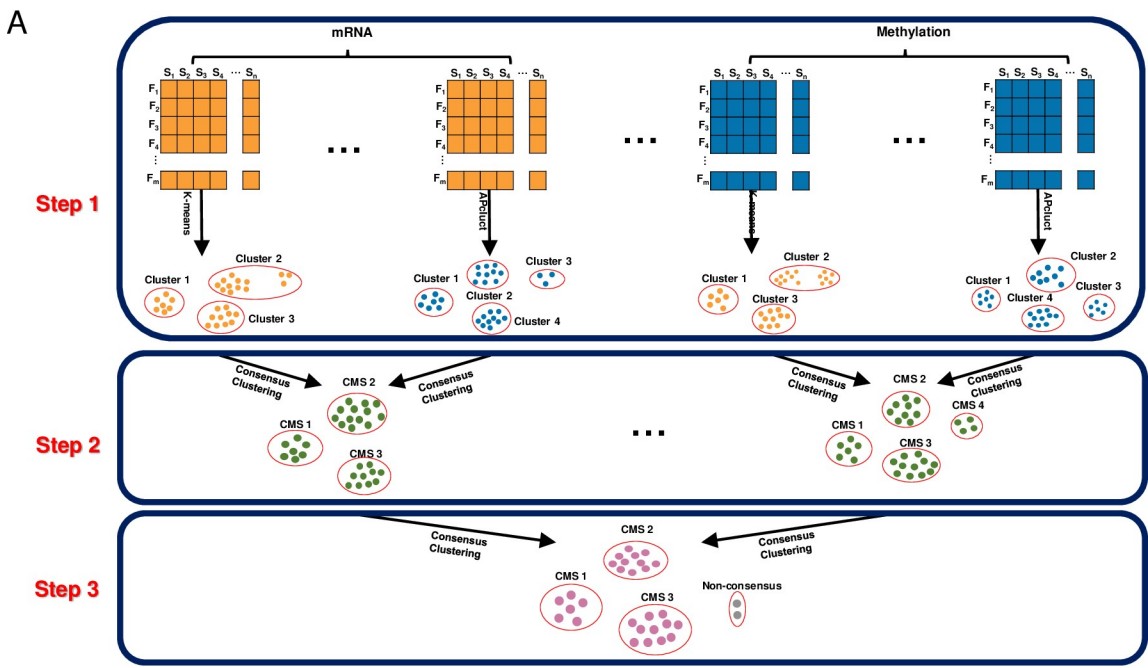

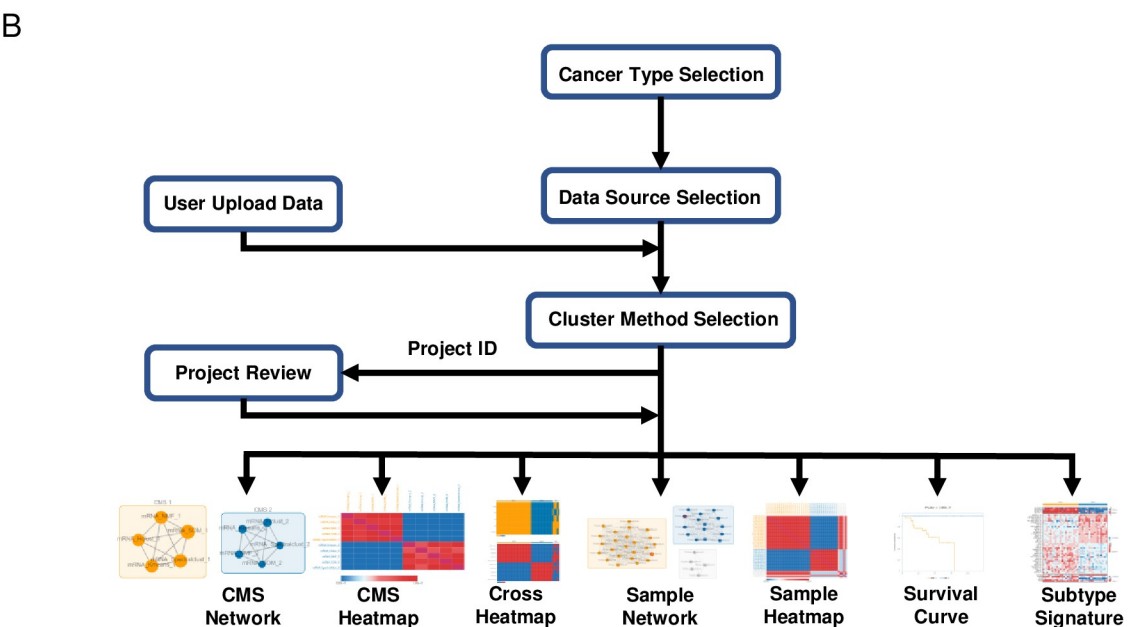

**Fig 1. Overview of the COMSUC.** (A) COMSUC workflow. (B) COMSUC user interface flowchart.

specific signatures. Notably, when an analysis includes users' data and reference public data, users' samples are highlighted in the network. All results can be downloaded and accessed for 30 days using a unique project ID and web link. Users can share the results with cooperators using the unique project ID.

A more detailed guide to the web server and its step-by-step use for basic analysis is provided in Supporting information (S1 Video) and on the "Manual" webpage of COMSUC.

## Data input

To upload private data for analysis, users click on "Data Source/Add File" on the "Analysis" page and select which omics data they want to upload. Each dataset should be uploaded as a tabular ".txt" file. Users can click on "Load Example Data" to automatically load all types of omics data. By clicking on "Download Example Data", users can download example data and confirm the data formatting requirements. If users want to upload their own data, they should upload at least one type of omics data and corresponding phenotype data. Both omics data and phenotype data file should be a "feature symbol (row) by sample name (column)" matrix. More format requirements for each type of omics data and phenotype data are detailed on the "Manual" page of the website. Notably, phenotype data are required. Sample names should be completely consistent across data files, or the web server will report an error.

## Analysis results

After submitting projects, users can discover cancer CMSs across clustering methods and omics data. COMSUC provides seven result types, which can be modified interactively for better presentation.

In the CMS network, each node corresponds to a subtype based on a single omics data type or method. Nodes are color coded according to their CMSs. Node size corresponds to the subtype sample number. Weighted edges encode Jaccard similarity coefficients between nodes; the width corresponds to the coefficient, and the transparency corresponds to its–log10 $P$ value. Users can change the layout and background color of the CMS network, cut edges by setting a threshold in the slide strip, and change other node and edge properties. Clicking on a node brings up information on that node in the right sidebar.

On CMS heatmaps, sample names are color coded according to CMSs. Users can change the color of the heatmap in the "Control Panel/Options." Selection of a heatmap block brings up sample information for that block in the right "Control Panel/Sample Information" section.

The cross heatmap illustrates per-sample distribution across each omics dataset or cluster method, grouped by CMSs and non-consensus groups. Columns represent samples and rows represent single omics data types or methods. Heatmap units represent the sample subtype divided based on single omics data types or methods. By clicking the heatmap, users can display another type of cross heatmap.

The sample network is a network of cancer CMSs across all samples. Each node represents a patient sample. Network edges correspond to highly concordant subtyping calls between samples. Nodes are color coded according to their CMS labels, with non-consensus samples displayed in gray. Users can interactively change the sample network as the same as CMS network. Users' samples are highlighted in the network.

On sample heatmaps, sample names are color coded according to CMSs. Users can also modify these heatmaps interactively.

Survival curves provide overall $P$ values and those for all pairs of CMSs. These values help users to evaluate the performance of consensus clustering.

Subtype signatures display the important features of each omics data type for CMS identification.

## Download data and results

There are three subpages on the "Download" page of the COMSUC website. All results can be downloaded in multiple formats on the "Download Analysis Results" subpage.

The public reference data after preprocessing and feature selection for 35 cancers are available for download on the "Download Sample Data (Feature selected)" subpage.

Moreover, if users want to preprocess multi-omics data of these cancers by themselves, they could also turn to "Download Sample Data" subpage to download data without preprocessing and feature selection. And then they could upload preprocessed data as private data to perform consensus clustering analysis in COMSUC.

### Example use case

To verify the performance of COMSUC, we implemented "Example 1", which is demonstrated in the website. We selected MCL consensus clustering method, five clustering methods (Hclust, Kmeans, NMF, SOM, and Spectralclust) and three types of omics data (mRNA, DNA methylation, and RPPA) for 44 ACC patient samples. We used COMSUC to integrate the data and cluster the samples into two CMSs (CMS1, $n$ = 21; CMS2, $n$ = 12; non-consensus, $n$ = 6; Fig 2A). Important signatures for the omics data are illustrated in Fig 2B. According to the Kaplan–Meier curve, the survival time for patients with ACC differed significantly between CMSs ($P$ = 0.00186; Fig 2C).

To illustrate to what degree did COMSUC performs better with integration of clustering methods, we compared the clustering results of single methods and consensus clustering method based on mRNA data of ACC, which is demonstrated in the website as "Example 2". The CMS heatmap illustrates that two subgroups generated by Kmeans and Hclust methods are the most dissimilar. The Jaccard similarity score between these two methods is 0.828 (Fig 2D). The cross heatmap shows that 5 non-consensus samples are divided into totally different subgroups by Kmeans and Hclust methods (Fig 2E). The $P$ values of survival analysis generated by these five clustering methods and consensus method are 0.000792 (for Hclust), 0.0150 (for Kmeans), 0.0150 (for NMF), 0.00130 (for SOM), 0.00340 (for Spectralclust), 0.00186 (for MCL consensus clustering), respectively. It suggests that different clustering methods generate discrepant clustering results and consensus clustering method generates more significant $P$ value of survival analysis than most of single clustering methods.

### Availability and future directions

The increasing amounts of cancer samples and complex multi-omics data pose new challenges for cancer molecular subtype analysis and visualization. We developed a comprehensive, user-friendly analytical web tool for cancer consensus subtype discovery (http://comsuc.bioinforai.tech/, http://59.110.25.27/). The intuitive nature of COMUC inputs and outputs help to assure that clinical doctors and biologists without programing knowledge have access to analysis of their data. To our knowledge, COMSUC is the first web server that allows users to perform clustering ensemble studies with multi-omics data, multiple methods, and multiple platforms. COMSUC has the following advantages.

i. Wide range of potential users. COMSUC is suitable for clinicians, researchers, and anyone else who wants to discover cancer molecular subtypes. For example, clinicians can upload their private patient data and select one of the public cancer project data sources to aid identification of CMSs in their patients. Cancer researchers can analyze their patient data alone to discover consensus subgroups. Biological researchers can identify cancer CMSs, explore CMS-specific characterization based on any cancer project, and prioritize potential therapeutic targets for each CMS.

ii. Special design for clinical research. Clinicians who have omics data from even single patients can use COMSUC to identify patient CMSs in the network of other samples from TCGA, ICGC, or TARGET. Moreover, COMSUC provides molecular evidence for CMSs on the "Subtype Signature" subpage to help clinicians optimize therapeutic strategies.

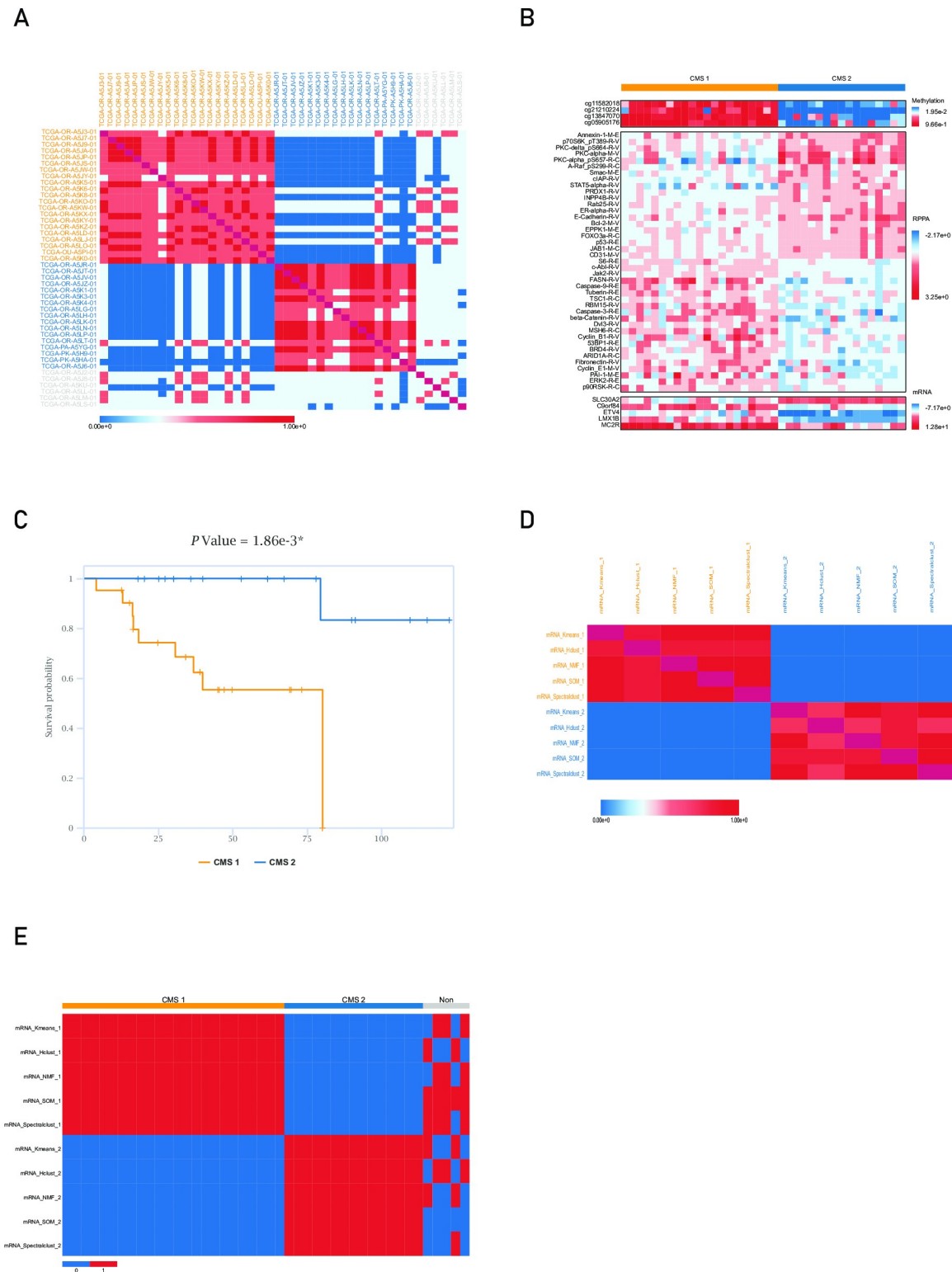

**Fig 2. The result of multiple methods and multi-omics consensus clustering.** (A) Sample heatmap of multi-omics consensus clustering. (B) Signatures that was important for CMSs identification. (C) Kaplan-Meier survival curve. (D) CMS heatmap of multiple methods consensus clustering. (E) Cross heatmap of multiple methods consensus clustering.

iii. Multiple typical algorithm selection. COMSUC provides eight optional algorithms with different characteristics and three consensus clustering methods that have been successfully applied to TCGA projects. Users can use all of these algorithms to generate a consensus CMS result, and evaluate the consistency of results based on single methods and consensus results on the "Cross Heatmap" subpage.

iv. Visualized, interactive, and publishable output of analysis results. COMSUC provides a step-by-step guide and video tutorial for users to submit their analysis projects. On the results page, users can interactively modify the networks, heatmaps, survival curves, and CMS-specific signature illustrations for better presentation. The CMS network and sample network presented by COMSUC can be saved as images in.PNG format, and exported as graphML files for analysis using Gephi. Heatmaps, survival curves, and subtype signatures can be saved as images in.PNG and.SVG formats.

## Supporting information

**S1 Video. A Video summary for COMSUC web server.**
(MP4)

**S1 File. Text for video summary.**
(DOCX)

**S2 File. Evaluation indexes of optimal cluster number.**
(DOCX)

## Author Contributions

**Conceptualization:** Song He, Xiaochen Bo.

**Data curation:** Xiaoxi Yang.

**Formal analysis:** Xiaoxi Yang.

**Funding acquisition:** Xiaochen Bo.

**Investigation:** Xinyu Song, Jijun Yu.

**Methodology:** Song He, Jijun Yu.

**Project administration:** Song He, Xinyu Song.

**Resources:** Song He, Jiannan Feng.

**Software:** Jijun Yu, Yuqi Wen, Lianlian Wu.

**Supervision:** Song He, Jiannan Feng, Xiaochen Bo.

**Validation:** Xinyu Song, Lianlian Wu, Bowei Yan.

**Visualization:** Xinyu Song, Jijun Yu, Yuqi Wen, Bowei Yan.

**Writing – original draft:** Xinyu Song, Xiaoxi Yang.

**Writing – review & editing:** Song He, Xiaoxi Yang, Xiaochen Bo.

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
