## [Decision Letter · Decision Letter 0]

28 Oct 2020

Dear Dr. Bo,

Thank you very much for submitting your manuscript "COMSUC: a web server for the identification of consensus molecular subtypes of cancer based on multiple methods and multi-omics data" for consideration at PLOS Computational Biology.

As with all papers reviewed by the journal, your manuscript was reviewed by members of the editorial board and by several independent reviewers. In light of the reviews (below this email), we would like to invite the resubmission of a significantly-revised version that takes into account the reviewers' comments.

We cannot make any decision about publication until we have seen the revised manuscript and your response to the reviewers' comments. Your revised manuscript is also likely to be sent to reviewers for further evaluation.

Sincerely,

Manja Marz

Software Editor

PLOS Computational Biology

Reviewer's Responses to Questions

**Comments to the Authors:**

Reviewer #1: In this manuscript the authors reported a web server, named COMSUC, used to identify the cancer Consensus Molecular Subtypes (CMSs). This tool was developed to integrate eight clustering methods and five types of omics data for 30 cancers, and was shown as being more user-friendly than previous similar tools. I noted that this lab has developed a web server ICM with integration of multi-omics data in 2016, so the current work seems to be an advance for the authors to address a need of more complex multi-omics data and more effective clustering methods. In view of the growth of omics data for cancer, I believe this work is important when the tool is extended based on more reasonable methods and designed with user-friendly operability. Herein, I have several points of concern on the manuscript and the COMSUC tool.

Major points

1. The authors emphasized several times that different clustering methods could lead to “discrepant” clustering results. This statement seems to be a little too vague. I would like to learn about what the discrepancy is among eight clustering methods? And, when they used COMSUC, to what degree did their strategy perform better (as well as being more reasonable) with integration of clustering methods? To answer these questions, perhaps some specific examples may be added into the subsection “Example use case”.

2. Moreover, in the subsection “Clustering algorithm”, the authors reported that they evaluated the optimal clustering number using different clustering performance assessment indexes. They listed these indexes according to their corresponding methods. However, I would like to learn about an important detail. How to assess the performance of a given solution of clustering in general? What about these indexes may play their roles in assessing the total performance of clustering? Maybe the authors should add a systematic analysis for this point.

3. As for multi-omics data used to analyze, it is certain that integration of omics data is helpful. The dataset contains five types of omics data from 14,954 patient samples: mRNA, miRNA, DNA methylation, copy number variation, and reverse-phase protein array (RPPA) data. The authors pointed out that the tool “… allows users to upload their own data for analysis”. Clearly it is difficult to ensure that the users’ private data are clean and good as these 14,954 patient samples. Could the authors provide an instruction of the data requirement for users’ private data? Moreover, for CMS identification, did these five types of omics data all work well? Could the authors provide a suggestion?

Minor points:

1. Page 5, Line 105, it is the first time that “MCL” appears in the text, what does “MCL” mean?

2. Page 8, Line 171-172, there are similar for “Hclust”, …, “NMF”, “SOM” etc., the authors should give their full names also. For example, “Hierarchical Clustering (Hclust),” …, “non-negative matrix factorization (NMF)”, “self-organizing maps (SOM)”

3. Page 6, Line128, should the format “.txt” be a tabular file?

Reviewer #2: The manuscript present an interesting idea for a webservice for computational discovery of molecular subtypes for diverse cancers. Thera are however some concerns that preclude me to express a clear (positive) recommendation for publication.

One important concern is usability. The main purpose of the manuscript is to advertise and introduce a service. However, the web service is UNAVAILABLE at both the URL and IP addresses. Since the methods are fairly standard and the data is publicly available. User experience would be a way to establish the merit of the work. This however cannot be made with unreachable websites.

The supplementary video seems fine, but does not replace user evaluation of the tool.

Aside from this, other concerns are related to documenting data processing (and pre-processing) more thoroughly since different omics call for different normalization and thresholding procedures that will impact clustering and consensus calculations. The thresholds provided "out of the blue" (disregarding sample sizes of the different databases, variability of the datasets and the different dynamic ranges of the omics) are misleading. As a quantitative biologist, I do not trust in "apparently" hand-waving arguments for calculations that have well established methods. However, a clinician may do this, precisely for this fact, solid, well-documented pre-processing methods must be provided so that quantitative biologist may explore and clinicians and experimentalist may trust the reliability of the results.

It will be interesting to document also why did the authors choose the MCL algorithm over competing alternatives. A solid rationale, perhaps supported by benchmark tests is desirable.

Once these concerns have been addressed, I may express a more favorable opinion on this work.

**Have all data underlying the figures and results presented in the manuscript been provided?**

Reviewer #1: Yes

Reviewer #2: Yes

PLOS authors have the option to publish the peer review history of their article (what does this mean?). If published, this will include your full peer review and any attached files.

Reviewer #1: **Yes: **Huaiqiu Zhu

Reviewer #2: **Yes: **Enrique Hernandez-Lemus
---

## [Decision Letter · Decision Letter 1]

31 Jan 2021

Dear Dr. Bo,

We are pleased to inform you that your manuscript 'COMSUC: a web server for the identification of consensus molecular subtypes of cancer based on multiple methods and multi-omics data' has been provisionally accepted for publication in PLOS Computational Biology.

Best regards,

Manja Marz

Software Editor

PLOS Computational Biology

Manja Marz

Software Editor

PLOS Computational Biology

Reviewer's Responses to Questions

**Comments to the Authors:**

Reviewer #1: With this revised manuscript, I saw a substantial revision in regard to my previous points. Herein I would like to thank the authors for their all responses to my comments, and totally, I am satisfied with the current revisions as well as their responses. A minor reminder should be pointed out is the affiliation writing for the authors, some places were described as “P.R. China” while some “China”, I think these should be written as the same way.

Reviewer #2: The authors have addressed my concerns

**Have all data underlying the figures and results presented in the manuscript been provided?**

Reviewer #1: Yes

Reviewer #2: Yes

PLOS authors have the option to publish the peer review history of their article (what does this mean?). If published, this will include your full peer review and any attached files.

Reviewer #1: **Yes: **Huaiqiu Zhu, PhD, Professor

Department of Biomedical Engineering, Peking University

Reviewer #2: **Yes: **Enrique Hernández-Lemus

---

## [Editor Report · Acceptance letter]

2 Mar 2021

PCOMPBIOL-D-20-01131R1 

COMSUC: a web server for the identification of consensus molecular subtypes of cancer based on multiple methods and multi-omics data

Dear Dr Bo,

I am pleased to inform you that your manuscript has been formally accepted for publication in PLOS Computational Biology. Your manuscript is now with our production department and you will be notified of the publication date in due course.

With kind regards,

Alice Ellingham
